# PGFinder, a novel analysis pipeline for the consistent, reproducible, and high-resolution structural analysis of bacterial peptidoglycans

Ankur V Patel[1]*, Robert D Turner[2], Aline Rifflet[3,4,5], Adelina E Acosta-Martin[6], Andrew Nichols[7], Milena M Awad[8], Dena Lyras[8,9], Ivo Gomperts Boneca[3,4,5], Marshall Bern[7], Mark O Collins[1,6]*, Stéphane Mesnage[1]*

[1]School of Biosciences, University of Sheffield, Sheffield, United Kingdom; [2]Department of Computer Science, University of Sheffield, Sheffield, United Kingdom; [3]Institut Pasteur, Unité Biologie et Génétique de la Paroi Bactérienne, Paris, France; [4]INSERM, Équipe Avenir, Paris, France; [5]CNRS, UMR 2001 "Microbiologie intégrative et moléculaire", Paris, France; [6]biOMICS Facility, Faculty of Science Mass Spectrometry Centre, University of Sheffield, Sheffield, United Kingdom; [7]Protein Metrics Inc, Cupertino, United States; [8]Infection and Immunity Program, Monash Biomedicine Discovery Institute, Clayton, Australia; [9]Department of Microbiology, Monash University, Clayton, Australia

*For correspondence:
apatel19@sheffield.ac.uk (AVP);
mark.collins@sheffield.ac.uk
(MOC);
s.mesnage@sheffield.ac.uk (SM)

**Abstract** Many software solutions are available for proteomics and glycomics studies, but none are ideal for the structural analysis of peptidoglycan (PG), the essential and major component of bacterial cell envelopes. It icomprises glycan chains and peptide stems, both containing unusual amino acids and sugars. This has forced the field to rely on manual analysis approaches, which are time-consuming, labour-intensive, and prone to error. The lack of automated tools has hampered the ability to perform high-throughput analyses and prevented the adoption of a standard methodology. Here, we describe a novel tool called PGFinder for the analysis of PG structure and demonstrate that it represents a powerful tool to quantify PG fragments and discover novel structural features. Our analysis workflow, which relies on open-access tools, is a breakthrough towards a consistent and reproducible analysis of bacterial PGs. It represents a significant advance towards peptidoglycomics as a full-fledged discipline.

## Introduction

The characterisation of bacterial cell walls started with the development of electron microscopy techniques (*Mudd and Lackman, 1941*), and it has ever since been the focus of countless studies. The major and essential component of the bacterial cell envelope is called peptidoglycan (PG). It confers cell shape and resistance to osmotic stress and represents an unmatched target for antibiotics (*Mainardi et al., 2008*; *Vollmer et al., 2008*). Some of the most widely used antibiotics to date (beta-lactams and glycopeptides) inhibit the polymerisation of PG.

PG (murein; originally known as mucopeptide) is a giant, insoluble, bag-shaped molecule, and its composition was characterised soon after its discovery (*Cummins and Harris, 1956*; *Rogers and Perkins, 1959*; *Weidel and Pelzer, 1964*). It is composed of glycan chains containing alternating *N*-acetylglucosamine (GlcNAc) and *N*-acetylmuramic acid (MurNAc) residues linked by β,1–4 bonds. The lactyl group of MurNAc residues is substituted by pentapeptide stems which often has

the L-Ala$_1$-γ-D-Glu$_2$-L-DAA$_3$-D-Ala$_4$-D-Ala$_5$ sequence, where DAA is a diamino acid such as *meso*-diaminopimelic (mDAP) acid or L-lysine (*Figure 1a*; *Vollmer et al., 2008*). In some species, a lateral chain (with variable composition and length) can be found attached to the amino acid in position 3. Peptide stem composition and polymerisation can vary amongst bacterial species (*Schleifer and Kandler, 1972*). Whilst PG building blocks produced in the cytoplasm are always the same, the final structure undergoes constant lysis and modification, a process referred to as 'remodelling'. Both remodelling and alternative polymerisation modes (*Figure 1b*) lead to a considerable variation in PG structure during cell growth and division. PG structural plasticity plays a critical role for adaption to environmental conditions during host-pathogen interaction (*Boneca et al., 2007*; *Juan et al., 2018*) or to survive exposure to antibiotics (*Mainardi et al., 2008*).

PG material is straightforward to purify, but the structural analysis of this molecule is challenging and remains a time-consuming and labour-intensive process. The intact molecule must be broken down into soluble fragments by enzymatic digestion with a glycosyl hydrolase (lysozyme), and individual building blocks (disaccharide peptides, also called muropeptides) are analysed to gain insight into the structure of the intact molecule. A transformative step for the characterisation of disaccharide peptides has been the use of reversed-phase HPLC (rp-HPLC) and mass spectrometry (MS) towards the end of the 1990s (*Garcia-Bustos et al., 1988*; *Glauner, 1988*; *Glauner et al., 1988*; *Martin et al., 1987*). Combining muropeptides separation by rp-HPLC and MS characterisation has hinted at a more complex structure than previously reported.

Despite tremendous advances in both rp-HPLC-MS instrumentation and software development for the automated analysis of large datasets, 'peptidoglycomics' is still in its infancy. The experimental strategy to analyse PG structure has barely changed over the past 30 years. Even though rp-HPLC-MS has been routinely used over the past decade, the analysis of MS data remains a black box. Except for a recent study describing the PG structure of *Pseudomonas aeruginosa* (*Anderson et al., 2020b*), no information is available in the literature about the strategy used to identify muropeptides in rp-HPLC-MS datasets. This task relies on searching a subset of expected structures, but the complexity of both the search space and search process is often not described.

We previously provided the proof of concept that shotgun proteomics tools can be used for the automated and unbiased analysis of PG structure (*Bern et al., 2017*). The analysis of *Clostridioides* (previously *Clostridium*) *difficile* PG led to the identification of many muropeptides never reported before. This work also demonstrated that PG analysis could be carried out with relatively high throughput, opening the possibility to analyse large numbers of samples (such as clinical or environmental isolates) using a minimal amount of material (typically microgram amounts).

Here, we describe a novel software called PGFinder for the analysis of MS data. PGFinder is a versatile and straightforward open-source software tool that allows automated identification of muropeptides based on the creation of dynamic databases. Sharing PGFinder as a Jupyter Notebook provides a robust and consistent pipeline with the potential to accelerate discovery in the field of peptidoglycomics. This workflow described here allows a comprehensive description of the analysis strategy for a consistent and reproducible PG structure analysis by users in the community.

We applied the PGFinder pipeline to analyse the muropeptides composition of *Escherichia coli* which has been extensively studied. We demonstrate that PGFinder can capture an unprecedented level of complexity of PG structure, highlighting the limitations of the search strategies reported so far. Finally, we provide evidence that PGFinder can be used in conjunction with freely available MS data deconvolution software, making PG analysis possible using entirely open-access tools. We propose that our approach represents a significant advance towards a consistent and reproducible analysis of PG structure, allowing peptidoglycomics to take the crucial first leap to parity with other omics disciplines.

## Results
### PGFinder: a dedicated script for bottom-up identification of PG fragments

No pipeline is currently available for the automated analysis of MS PG data. Therefore, we sought to replicate a shotgun proteomics approach to create an analysis pipeline dedicated to PG analysis, referred to as 'peptidoglycomics' (*Wheeler et al., 2014*).

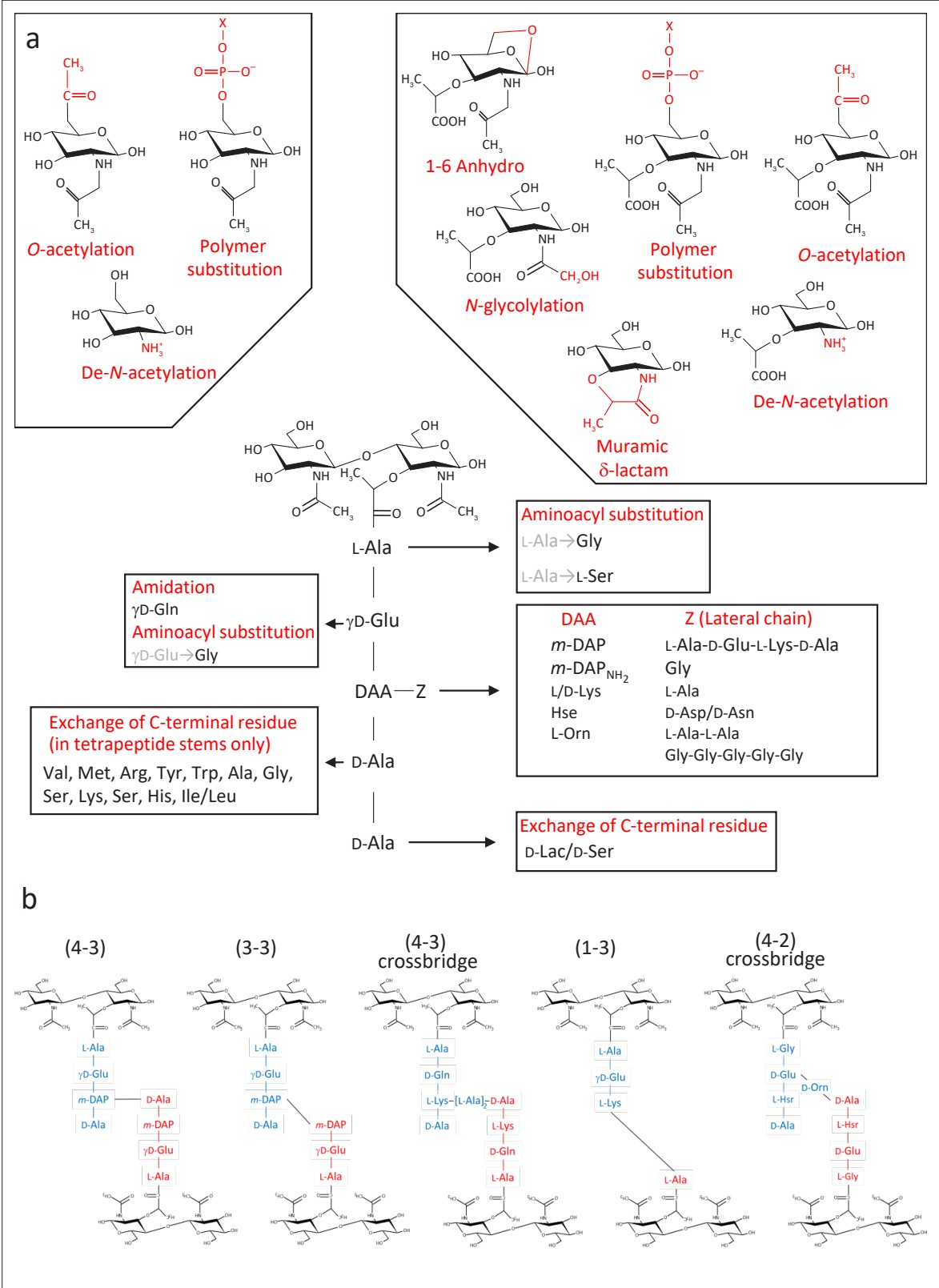

**Figure 1.** Diversity of peptidoglycan composition and structure. (**a**) Representative peptidoglycan building block made of *N*-acetylglucosamine (GlcNAc) and *N*-acetylmuramic acid (MurNAc) forming a disaccharide subunit linked to a pentapeptide stem attached to the MurNAc via a lactyl moiety. Peptide stem contains both L and D-amino acids and show a great diversity in composition. Some examples of amino acids found in peptidoglycan are shown for each residue. Modifications of the sugars are also shown. (**b**) Representation of crosslinking diversity, 4–3 bonds (direct or via peptide

*Figure 1 continued on next page*

*Figure 1 continued*

crossbridge) and 3–3 bonds are made by D,D- or L,D-transpeptidases, respectively. The enzymes catalysing 1–3 and 4–2 bonds remain unknown. Acceptors stems are shown in blue and donor stems in red. DAA: diamino acid; *m*-DAP: *meso*-diaminopimelic acid; D-Lac: D-lactate; X: cell surface polymer (e.g teichoic acid); Z: lateral chain.

To limit misidentifications due to mass coincidences, we established a search strategy relying on an iterative process (*Figure 2*). A first search was carried out using a database made of reduced disaccharide peptides (monomers) and their theoretical monoisotopic masses. MS data were deconvoluted using the Protein Metrics Byos software to generate a list of observed monoisotopic masses alongside other parameters including retention times and signal intensity. Individual theoretical masses contained in the monomer database (*Figure 2*, database 1) were compared with observed masses in the experimental dataset. Any observed mass within 10 ppm tolerance was considered as a match and the corresponding inferred structure and theoretical mass were then added to a list of matched structures (*Figure 2*, library 1). As a second step, we used the list of matched monomers to build another database in silico (*Figure 2*, database 2), corresponding to dimers and trimers and their theoretical masses. Two types of polymerisation events are included in the original PGFinder version depending on the type of crosslink either through peptide stems or glycan chains. Individual theoretical masses from the in silico database were compared to observed masses to generate a list of matched dimers and trimers (*Figure 2*, library 2). As a third step, we combined the lists of matched monomers and multimers to generate a final library of modified muropeptides (*Figure 2*, library 3). The final library contained only modified muropeptides corresponding to matched monomers, dimers, and trimers. The modifications accounted for include the presence of anhydro groups, deacetylated sugars, amidated amino acids, and modifications resulting from *N*-acetylglucosaminidase or amidase activities (loss of GlcNAc and lack of peptide stems, respectively). In-source decay products (loss of GlcNAc) and $Na^+$/$K^+$ salt adducts were also added to library 3. All three libraries corresponding to observed monomers, dimers, trimers, and their modified variants were combined to search the MS data for masses matching theoretical values within a 10 ppm mass accuracy window. This search generated results processed by PGFinder to carry out a 'clean up step'. The intensities of in-source decay products and salt adducts were combined with that from parent ions when found within close

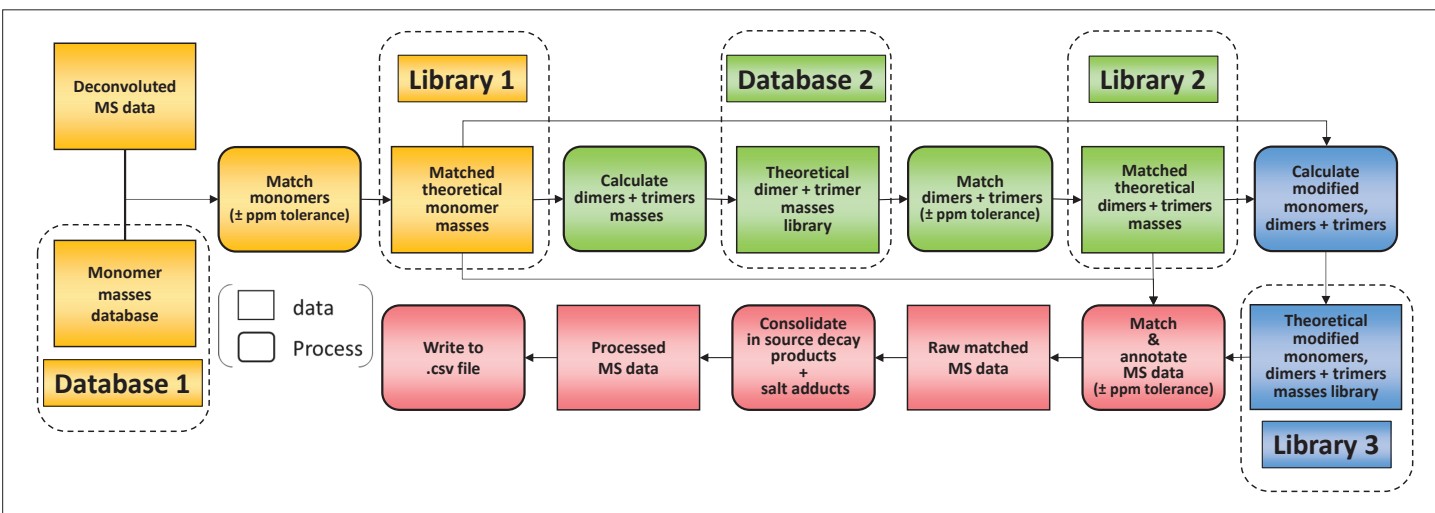

**Figure 2.** Flowchart outlining the algorithm for the matching script. The identification of muropeptides was carried out using four successive steps, indicated by different colours (orange, green, blue, and red, respectively). As a first step, observed masses in the dataset are compared to a list of theoretical masses corresponding to monomers (database 1). Matched masses within the ppm tolerance set (10 ppm for Orbitrap data) are used to build a list of inferred monomeric structures and their corresponding theoretical masses (library 1). This is then used to generate a list of theoretical multimers (dimers and trimers) and their masses (database 2). A second matching round is carried out to build a list of inferred multimers (library 2). At this stage, matched monomers and multimers are combined to generate a list of modified muropeptides (library 3). Two libraries of matched theoretical masses (monomers and dimers, trimers) and a third library (their modified counterparts) are used to search the dataset. Muropeptide structures are inferred from a match within tolerance between theoretical and observed masses. This data is then 'cleaned up' by combining the intensities of ions corresponding to in-source decay and salt adducts to those of parent ions. The final matched mass spectrometry data is then written to a .csv file.

retention time (a 0.5 min time window). The output of this final step is a matched table written to a .csv format file. It contained all the inferred structures identified within the specified mass and retention time windows with an extracted-ion chromatogram (XIC) signal intensity for quantification.

## Using PGFinder to investigate PG structure and identify low-abundance muropeptides

The performance of the matching script was tested using the well-characterised PG from *E. coli* as a proof of concept. UHPLC-MS/MS data were acquired for three independent PG samples (biological replicates; *Appendix 1—figure 1*). Following MS1 spectral deconvolution (a process calculating masses from observed *m/z* values), observed masses were matched to theoretical muropeptides masses according to the strategy described above (*Figure 2*). A first search was carried out using a minimal mass database made of 10 simple PG fragments including three glycan chains (di-, tetra-, and hexasaccharides) and seven monomers (*Table 1—source data 1*). Due to the .csv format of the database, the diamino acid in position 3 could not be assigned to a symbol or Greek letter and had to be one of the 26 letters already assigned by the IUPAC-IUB Joint Commission on Biochemical Nomenclature. We used the letter J for mDAP for the initial search and replaced it by the letter m in the final table. The output of the automated search is a .csv file per dataset; all files corresponding to biological replicates were collated into one Excel file (*Table 1—source data 2*). Each search output contained approximately 3000 rows of masses and corresponding parameters. Depending on the dataset analysed, 41–48% of the total ion intensity was assigned to PG structures. As anticipated, inferred structures were frequently found with multiple retention times, reflecting the existence of stereoisomers, with one species accounting for most of the intensity. In some cases, observed masses matched with more than one inferred structure. The output of the automated search was consolidated as described in *Supplementary file 1*. Retention times were assigned to individual structures based on the elution of the most abundant stereoisomer. For example, >97% of the most abundant monomer GM-AEJA was eluted at an average retention time of 10.04 ± 0.04 min. Data consolidation revealed an unprecedented muropeptide composition complexity compared to recent LC-MS analyses of *E. coli* (*Kühner et al., 2014*; *Table 1—source data 2*). Sixty PG fragments were identified (*Table 1*): these included glycan chains lacking peptide stems (4.38%), monomers (63.14%), dimers (29.54%), and trimers (2.94%) (*Figure 3*). Based on the abundance of multimeric PG fragments, we report a crosslinking index of 15.69%, which is slightly lower than the value previously reported of 23.1% (*Glauner, 1988*).

The automated and unbiased search revealed several muropeptides that were expected but never reported to date for *E. coli*. These included (i) PG fragments resulting from amidase activity (4.55%), found as 'denuded glycans' (disaccharides and tetrasaccharides) and modified variants or muropeptide stem with an extra disaccharide residue; (ii) a low-abundance (0.23%) PG fragments containing deacetylated GlcNAc residues; and (iii) PG fragments resulting from glucosaminidase activity (0.12%). Deacetylated muropeptides were not expected since no *E. coli* PG deacetylase has been identified in this organism to date. All the structures identified for the first time (glycan chains, monomers containing deacetyl groups, and muropeptides lacking a GlcNAc residue) were confirmed by MS/MS analysis (*Table 1—source data 3*). The proportion of muropeptides with anhydroMurNAc groups identified (4.55%) was in line with previous studies, yielding an average chain length of 36.05. This value is higher than an earlier study that reported a predominant chain length of 5–10 disaccharide units (*Harz et al., 1990*), but agreed with recent work that reported long glycan chains in *E. coli* (*Turner et al., 2018*). Overall, the quantification of muropeptides across biological replicates was very consistent, with Pearson's correlation coefficients >0.96 (*Appendix 1—figure 2*). The most pronounced variations in quantification were observed with low-abundance muropeptides accounting for less than 1% of the species identified.

We further explored the structural diversity of *E. coli* PG, performing a more complex search with a mass database made of glycan chains and all possible monomers containing di-, tri-, tetra-, and penta-peptide stems (*Table 1—source data 4*; 224 structures in total). Several monomers with tetra- and pentapeptide stems containing unusual amino acids were identified. Only four structures could be confirmed by MS/MS analysis and corresponding multimers were retained (GM-AEJF, -AEJN, AEJK, -AEJAK, and GM-AEJKD). Collectively, muropeptides containing unusual amino acids accounted for ca. 7.5% of the 80 structures identified (*Table 1—source data 5*) using a complex mass database.

**Table 1.** Processed match output.

| | Structure | RT (min) Av±SD | Abundance (%) Av±SD | Monoisotopic mass (Da) Obs | Theo | Δppm |
|---|---|---|---|---|---|---|
| | GM\|0 | 3.62±0.01 | 3.465±0.683 | 498.205 | 498.206 | 2.5 |
| Glycans | GM (x2)\|0 | 10.11±0.03 | 0.428±0.349 | 976.384 | 976.386 | 2.2 |
| 4.38%±0.35% | GM (anhydro) \|0 | 8.20±1.92 | 0.238±0.025 | 478.179 | 478.180 | 2.9 |
| | GM (deacetyl) \|0 | 2.57±0.00 | 0.155±0.032 | 456.194 | 456.196 | 3.5 |
| | GM (x2) (deacetyl) \|0 | 6.86±0.02 | 0.093±0.012 | 934.372 | 934.376 | 3.2 |
| | GM-AEmA\|1 | 10.04±0.04 | 36.098±2.131 | 941.405 | 941.408 | 2.8 |
| | GM-AEm\|1 | 6.57±0.01 | 14.352±0.397 | 870.368 | 870.371 | 3.0 |
| | GM-AEmKR\|1 | 9.56±0.05 | 8.030±0.774 | 1154.563 | 1154.567 | 3.6 |
| | GM-AE\|1 | 9.57±0.04 | 1.809±0.231 | 698.284 | 698.286 | 3.1 |
| | GM-AEmG\|1 | 7.85±0.05 | 0.689±0.049 | 927.390 | 927.392 | 2.3 |
| | GM-AEm (anhydro) \|1 | 13.98±0.02 | 0.668±0.073 | 850.342 | 850.344 | 2.2 |
| Monomers | GM-AEmA (anhydro) \|1 | 16.55±0.01 | 0.573±0.100 | 921.380 | 921.382 | 2.0 |
| 63.14%±1.13% | GM-AEmAG\|1 | 9.45±0.05 | 0.219±0.009 | 998.426 | 998.429 | 3.1 |
| | GM-AEmKR (anhydro) \|1 | 14.83±0.01 | 0.160±0.039 | 1134.537 | 1134.540 | 2.9 |
| | GM-AEmA (deacetyl) \|1 | 8.57±0.06 | 0.083±0.055 | 899.394 | 899.397 | 3.1 |
| | GM-GM-AEmA\|1 | 13.10±0.02 | 0.075±0.040 | 1419.584 | 1419.588 | 2.9 |
| | GM-AE (anhydro) \|1 | 17.44±0.01 | 0.069±0.013 | 678.258 | 678.260 | 2.8 |
| | M-AEm\|1 | 4.56±0.01 | 0.062±0.064 | 667.289 | 667.291 | 3.8 |
| | M-AEmKR\|1 | 8.16±0.06 | 0.061±0.056* | 951.484 | 951.487 | 3.2 |
| | GM-AEmAA\|1 | 11.38±0.04 | 0.059±0.003 | 1012.442 | 1012.445 | 2.4 |
| | M-AEmA\|1 | 8.52±0.05 | 0.053±0.015 | 738.325 | 738.328 | 4.0 |
| | GM-GM-AEm\|1 | 11.31±0.04 | 0.042±0.025 | 1348.547 | 1348.551 | 2.4 |
| | GM-AEm (deacetyl) \|1 | 4.77±0.01 | 0.024±0.014 | 828.358 | 828.360 | 3.0 |
| | GM-GM-AEmKR\|1 | 12.18±0.03 | 0.011±0.002* | 1632.742 | 1632.747 | 3.0 |
| | GM-AEmA-GM-AEmA\|2 | 16.01±0.02 | 17.247±0.777 | 1864.800 | 1864.805 | 2.3 |
| | GM-AEmA-GM-AEmKR\|2 | 14.83±0.02 | 4.589±0.589 | 2077.957 | 2077.964 | 3.0 |
| | GM-AEmA-GM-AEm\|2 | 15.09±0.02 | 3.207±0.168 | 1793.763 | 1793.768 | 2.6 |
| | GM-AEmA-GM-AEmA (anhydro) \|2 | 20.56±0.01 | 0.873±0.037 | 1844.774 | 1844.778 | 2.4 |
| | GM-AEm-GM-AEmKR\|2 | 14.22±0.00 | 0.855±0.101 | 2006.920 | 2006.926 | 3.3 |
| | GM-AEmA-GM-AEmKR (anhydro) \|2 | 18.89±0.17 | 0.665±0.079 | 2057.934 | 2057.937 | 1.8 |
| | GM-AEm-GM-AEm\|2 | 14.23±0.01 | 0.558±0.062 | 1722.725 | 1722.730 | 3.0 |
| | GM-AEm-GM-AEmAG\|2 | 14.68±0.01 | 0.416±0.025 | 1850.785 | 1850.789 | 2.4 |
| | GM-AEmA-GM-AEm (anhydro) \|2 | 19.66±0.01 | 0.381±0.028 | 1773.738 | 1773.741 | 2.1 |
| Dimers | GM-AEmA-GM-AEmAG\|2 | 15.33±0.02 | 0.179±0.005 | 1921.822 | 1921.826 | 2.2 |
| 29.54%±0.46% | GM-AEm-GM-AEmKR (anhydro) \|2 | 18.07±0.01 | 0.170±0.024 | 1986.896 | 1986.900 | 2.1 |
| | GM-AEm-GM-AEm (anhydro) \|2 | 18.77±0.01 | 0.141±0.015 | 1702.697 | 1702.704 | 4.5 |
| | GM-AEmA-GM-AEmAA\|2 | 16.54±0.01 | 0.075±0.002 | 1935.838 | 1935.842 | 2.1 |

*Table 1 continued on next page*

*Table 1 continued*

| | Structure | RT (min) Av±SD | Abundance (%) Av±SD | Monoisotopic mass (Da) Obs | Theo | Δppm |
|---|---|---|---|---|---|---|
| | GM-AEm-GM-AEmG\|2 | 13.91±0.01 | 0.054±0.003 | 1779.747 | 1779.752 | 2.7 |
| | GM-GM-AEmA-GM-AEmA\|2 | 17.51±0.01 | 0.046±0.028 | 2342.976 | 2342.985 | 3.6 |
| | GM-AEmA-GM-AEmA (deacetyl) \|2 | 15.17±0.01 | 0.029±0.022 | 1822.789 | 1822.794 | 3.0 |
| | GM-AEmA-GM-AEmG (anhydro) \|2 | 19.12±0.01 | 0.021±0.001 | 1830.761 | 1830.763 | 0.7 |
| | GM-AEmA-GM-AEmAG (anhydro) \|2 | 19.73±0.01 | 0.019±0.002 | 1901.796 | 1901.800 | 2.1 |
| | GM-AEmA-GM-AEmAA (anhydro) \|2 | 21.17±0.02 | 0.015±0.002 | 1915.812 | 1915.816 | 1.8 |
| | GM-GM-AEmA-GM-AEm\|2 | 16.85±0.00 | 0.003±0.004 | 2271.943 | 2271.947 | 2.1 |
| | GM-AEmA-GM-AEmA-GM-AEmA\|3 | 18.86±0.01 | 1.751±0.221 | 2788.192 | 2788.202 | 3.5 |
| | GM-AEmA-GM-AEmA-GM-AEm\|3 | 18.23±0.21 | 0.371±0.031 | 2717.158 | 2717.164 | 2.2 |
| | GM-AEmA-GM-AEmA-GM-AEmA (anhydro) \|3 | 22.39±0.02 | 0.222±0.027 | 2768.169 | 2768.175 | 2.3 |
| | GM-AEmA-GM-AEmA-GM-AEmKR\|3 | 17.54±0.01 | 0.207±0.028 | 3001.350 | 3001.360 | 3.4 |
| | GM-AEmA-GM-AEmA-GM-AEm (anhydro) \|3 | 21.60±0.02 | 0.117±0.003 | 2697.133 | 2697.138 | 1.8 |
| Trimers | GM-AEmA-GM-AEmA-GM-AEmKR (anhydro) \|3 | 20.90±0.16 | 0.088±0.026 | 2981.328 | 2981.334 | 2.2 |
| 2.94%±0.36% | GM-AEmA-GM-AEmA-GM-AEmG\|3 | 17.72±0.01 | 0.039±0.004 | 2774.182 | 2774.186 | 1.4 |
| | GM-AEmA-GM-AEm-GM-AEm\|3 | 17.45±0.01 | 0.029±0.005 | 2646.123 | 2646.127 | 1.7 |
| | GM-AEmA-GM-AEm-GM-AEm (anhydro) \|3 | 21.16±0.01 | 0.025±0.001 | 2626.096 | 2626.101 | 1.9 |
| | GM-AEmA-GM-AEm-GM-AEmKR\|3 | 17.11±0.01 | 0.022±0.002 | 2930.316 | 2930.323 | 2.7 |
| | GM-AEmA-GM-AEmA-GM-AEmAG\|3 | 18.24±0.01 | 0.021±0.001 | 2845.217 | 2845.223 | 2.0 |
| | GM-AEmA-GM-AEmA-GM-AEmAA\|3 | 19.23±0.01 | 0.014±0.002* | 2859.235 | 2859.239 | 1.3 |
| | GM-AEmA-GM-AEm-GM-AEmKR (anhydro) \|3 | 20.31±0.02 | 0.014±0.005 | 2910.293 | 2910.297 | 1.5 |
| | GM-AEm-GM-AEmG-GM-AEmAG\|3 | 17.18±0.00 | 0.004±0.005 | 2703.143 | 2703.149 | 2.0 |
| | GM-AEmA-GM-AEm-GM-AEmG (anhydro) \|3 | 21.21±0.02 | 0.011±0.003* | 2754.157 | 2754.160 | 1.1 |
| | GM-AEmA-GM-AEmA-GM-AEmAG (anhydro) \|3 | 21.77±0.01 | 0.006±0.004 | 2825.189 | 2825.197 | 2.8 |

Inferrred dimers and trimers are based on the most abundant monomers and could correspond to alternative structures.

G: GlcNAc; M: MurNAc; m: *meso*-diaminopimelic acid; the number following the symbol '|' refers to the oligomerisation state (1 for monomers, 2 for dimers, and 3 for trimers).

*Calculated from two values.

The online version of this article includes the following source data for table 1:

**Source data 1.** *E. coli* simple mass database.

**Source data 2.** *E. coli* matching output and consolidated data.

**Source data 3.** MS/MS analysis of *E. coli* glycan chains and monomers.

*Table 1 continued on next page*

Table 1 continued
**Source data 4.** *E. coli* complex mass database.
**Source data 5.** *E. coli* muropeptide complex table.

## Using PGFinder for the comparative analysis of PG structures

We showed with *E. coli* data that PGFinder is suitable to characterise the high-resolution structure of PGs using a 'bottom-up' approach. However, this requires a careful analysis of the search output to confirm the identity of muropeptides identified and discriminate between multiple structures that can be assigned to a unique observed mass. A more basic application is the use of PGFinder in organisms that have already been studied in detail to either compare PG composition or quantify the abundance of specific structures. This application accounts for most PG analyses described in the

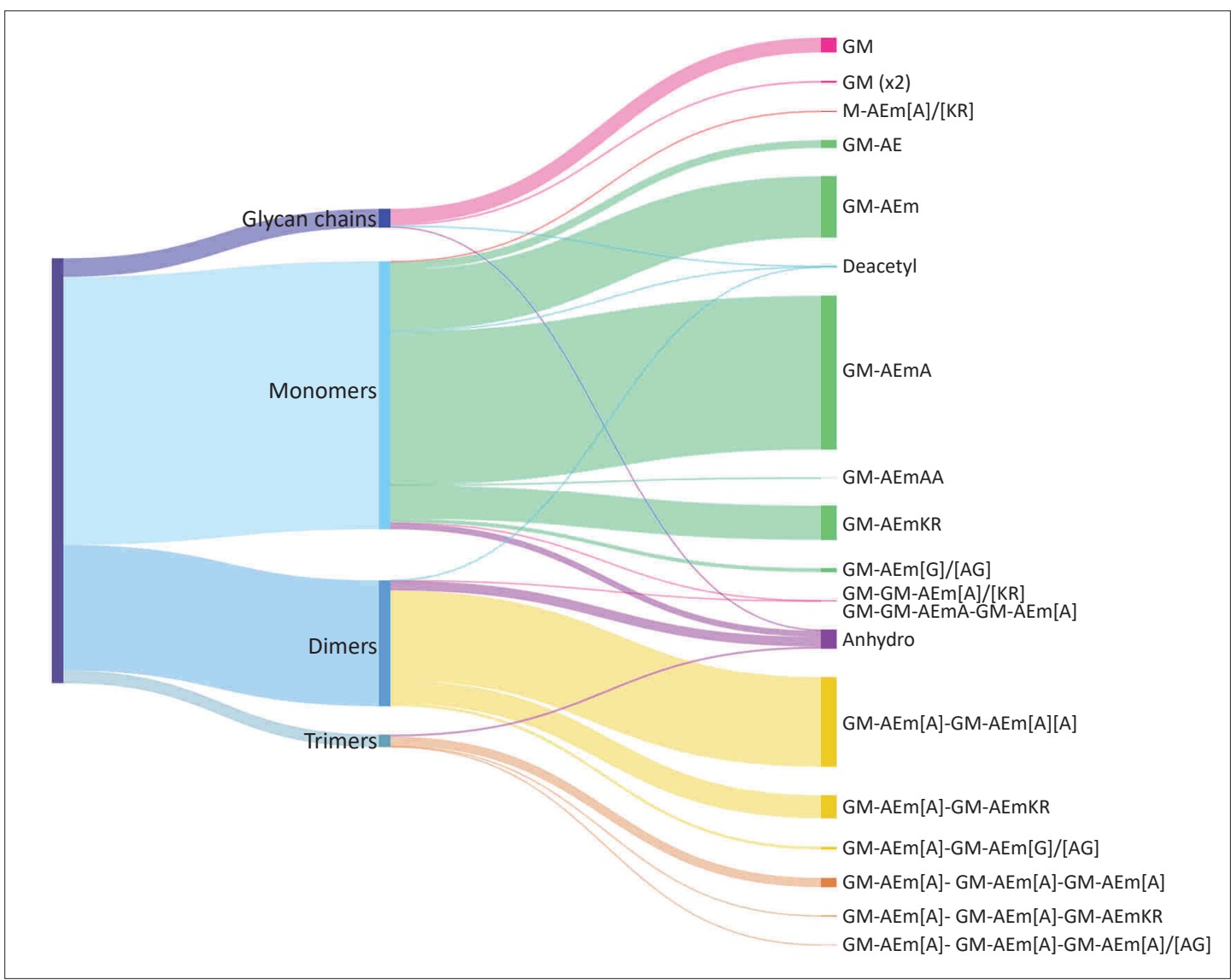

**Figure 3.** Distribution of *E. coli* peptidoglycan fragments identified using automated search workflow. Breakdown of peptidoglycan is shown by oligomerisation state (left) branching to specific composition (right). Branch size is proportional to percentage. Monomers, dimers, trimers, and glycan chains (left) are broken down into muropeptide composition and structure (right). Individual structures are grouped by colour according to oligomerisation state. Monomers, green; dimers, yellow; trimers, orange. Residues in square brackets are only found in some muropeptides. For example, GM-AEJ[A]-GM-AEJ[A] can represent GM-AEJA-GM-AEJA, GM-AEJA-GM-AEJ, and GM-AEJ-GM-AEJ. G: *N*-acetylglucosamine; M, *N*-acetylmuramic acid; A: L- or D- alanine; E: γ -D-glutamic acid; J: *meso*-diaminopimelic acid; K: D-lysine; R: D-arginine; G: glycine.

literature, comparing PG structures between different isolates, isogenic mutants, or cells grown in different conditions.

We chose the PG of *C. difficile* as a proof of concept to demonstrate how PGFinder can carry out a straightforward comparative analysis. We prepared PG samples corresponding to biological triplicates from two clinical isolates, R20291 and M7404. To illustrate the versatility of the software, PG samples were digested by mutanolysin, and disaccharide peptides were beta-eliminated to generate lactyl-peptides (*Tipper, 2002*) and analysed by UHPLC-MS (*Figure 4—figure supplement 1*). Since the high-resolution PG structure of *C. difficile* has been described based on the MS/MS analysis of muropeptides (*Bern et al., 2017*), we used data previously published to build a PG fragment database for a 'one-off' matching step. The database included monomers identified by MS/MS, containing unusual amino acids and the corresponding dimers resulting from either D,D- or L,D-transpeptidation (with a AEJA or AEJ peptide as donor stem). To limit the complexity of the search output, we limited the list of trimers, tetramers, and pentamers to those containing the most abundant peptide stems found in dimers (AEJA, AEJ, or AEJG). The database of theoretical masses contained 74 PG structures described in *Figure 4—source data 1*. MS data were deconvoluted using Byos Feature Finder, and observed monoisotopic masses were matched to theoretical masses using PGFinder. To perform the matching operation in its simplest form, all options offered by the software were deactivated.

All monomers and dimers searched were identified, except for two muropeptides containing a peptide stem with a methionine residue in position four, both present in low abundance in *C. difficile* strain 630 (ca. 0.10%). 25 out of the 31 possible trimers, tetramers, and pentamers searched were found (*Figure 4—source data 2*). Comparison of biological replicates revealed a high reproducibility, both in retention times and quantification. A high correlation was found between biological replicates, confirming the robustness of the quantification method (*Figure 4a*). Both strains contained a similar amount of mono-, tri-, tetra-, and pentamers, but strain M7404 contained a significantly lower proportion of monomers and a higher proportion of dimers than strain R20291 (p=0.022 and p=0.005, respectively; Student's *t*-test; *Figure 4b*). We next performed a Student's *t*-test using permutation-based FDR to identify statistically significant differences in the abundance of individual muropeptides between the two strains. The p-value was plotted on a volcano plot against the fold change in abundance between the two samples (*Figure 4c*). Two muropeptides were significantly less abundant (Lac-AEJ[AG] and Lac-AEJ-Lac-AEJA), and four others were significantly more abundant in strain R20291 (Lac-AEJV- Lac-AEJA, Lac-AEJ[L/I]-Lac-AEJA, Lac-AEJAA-Lac-AEJA and the trimer (Lac-AEJA)3). These differences are likely to reflect different substrate specificities for PBPs and the Ddl ligases in these strains. Therefore, combining the output of PGFinder with statistical analysis of muropeptide abundance offers a robust workflow to identify differences in PG composition.

## Benchmarking the automated PG analysis pipeline using available datasets

The analysis of *E. coli* PG established a proof of concept, showing that our matching script is suitable for the automated analysis of PG MS data. We next sought to evaluate the robustness of our peptidoglycomics pipeline using available datasets described in the literature. The most suitable publication was a recent study by Anderson et al. describing a PG analysis of *P. aeruginosa* planktonic cells (*Anderson et al., 2020b*). Unlike most (if not all) studies published to date, this work provided datasets from biological and technical triplicates. Unlike our *E. coli* samples, analysed on a Q Exactive Focus Orbitrap (Thermo), *P. aeruginosa* samples were analysed on an Agilent Q-TOF mass spectrometer. Spectra were deconvoluted using the Byos Feature Finder module, and observed masses were matched using a mass tolerance of 25 ppm as described by Anderson et al. To limit the occurrence of mass coincidences and misidentification at this slightly lower mass accuracy, we carried out a search with PG modifications (anhydroMurNAc residues, deacetylation, lack of peptide stem resulting from amidase activity and amidation) but only using the most frequent combinations of modifications (double anhydroMurNAc and anhydroMurNAc and deacetyl).

PGFinder identified 63 muropeptides out of the 71 reported by Anderson et al., matching our search criteria (*Table 2*). The eight muropeptides that were not identified were absent from the list of deconvoluted masses, indicating that the problem was not associated with the script, highlighting that the deconvolution step is a source of variability. Interestingly, the observed masses calculated using Byos Feature Finder were closer to the theoretical value (6.5 ppm versus 10.7 ppm on average),

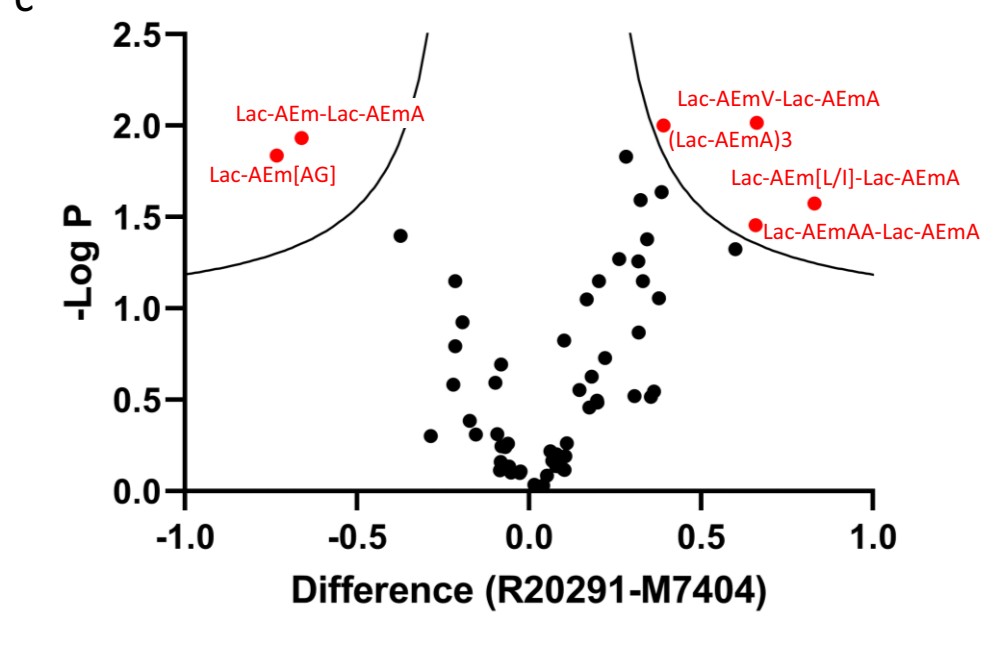

**a**

|  | R20291.1 | R20291.2 | R20291.3 | M7704.1 | M7704.2 | M7704.3 |
|---|---|---|---|---|---|---|
| **R20291.1** | NA | 0.996 | 0.995 | 0.994 | 0.991 | 0.994 |
| **R20291.2** | 0.996 | NA | 0.996 | 0.995 | 0.995 | 0.992 |
| **R20291.3** | 0.995 | 0.996 | NA | 0.994 | 0.993 | 0.989 |
| **M7704.1** | 0.994 | 0.995 | 0.994 | NA | 0.998 | 0.995 |
| **M7704.2** | 0.991 | 0.995 | 0.993 | 0.998 | NA | 0.991 |
| **M7704.3** | 0.994 | 0.992 | 0.989 | 0.995 | 0.991 | NA |

**b**

|  | R20291 | M7404 | *P* value |
|---|---|---|---|
| Monomers | 52.81% ± 2.09% | 46.17% ± 2.38% | 0.022 |
| Dimers | 36.12% ± 1.48% | 43.34% ± 1.74% | 0.005 |
| Trimers | 10.27% ± 0.64% | 9.73% ± 0.67% | NS |
| Tetramers | 0.72% ± 0.08% | 0.68% ± 0.02% | NS |
| Pentamers | 0.07% ± 0.00% | 0.06% ± 0.00% | NS |
| Crosslinking index | 21.64% ± 0.93% | 21.86% ± 0.87% | NS |

**c**

**Figure 4.** Comparative analysis of *C. difficile* R20291 and M7404 peptidoglycan (PG) composition. (**a**) Pearson's correlation coefficients across biological replicates of R20291 and M7404 *C. difficile* isolates. Heatmap gradient shows highest value in green to lowest value in red. (**b**) Muropeptide distribution according to degree of crosslinking. Comparison was carried out using a Student's *t*-test; p-value is indicated for each category of muropeptides. (**c**) Volcano plot, where each dot represents an individual muropeptide, plotted against the significance (Student's *t*-test p-value<0.05, FDR < 0.05, $S_0$ = 0.1) and difference ($\log_2$). Muropeptides showing a significantly different abundance between strains are highlighted in red. Lac: lactyl group; A: D/L-alanine; E: $\gamma$-D-glutamate; J: *meso*-diaminopimelic acid V: D-valine; L: D-leucine; I: D-isoleucine; G: glycine.

The online version of this article includes the following figure supplement(s) for figure 4:

**Source data 1.** *C. difficile* mass database.

*Figure 4 continued on next page*

*Figure 4 continued*

**Source data 2.** *C. difficile* 20291 versus M7404, list of muropeptides, abundance, RT.

**Figure supplement 1.** *C. difficile* LC-MS chromatograms.

reflecting another source of variability associated with data deconvolution. Four muropeptides previously identified containing the AEJAG pentapeptide stem were matched with distinct structures due to a mass coincidence between K and AG. A careful analysis of MS/MS spectra suggested that these muropeptides contained a K residue rather than the AG dipeptide (the y2 ion being 143 ppm away from the expected mass), showing the added value of an unbiased search. This conclusion is supported by the retention times of the corresponding muropeptides since the tetrapeptide AEJK elutes before EAJA whilst the pentapeptide AEJAG elutes later in the chromatography (*Bern et al., 2017*). It is worth noting that our search also identified a large number of muropeptides that were not reported previously (*Table 2—source data 1*). These results collectively show that our PG analysis pipeline can identify an unprecedentedly large number of muropeptide structures based on MS1 data, including all those previously reported (*Anderson et al., 2020a*; *Anderson et al., 2020b*).

## A PG analysis workflow using freely available tools

The automated identification of *P. aeruginosa* muropeptides using PGFinder and the strategy reported previously (*Anderson et al., 2020a*; *Anderson et al., 2020b*) relied on commercially available deconvolution software (ProteinMetrics Byos or Agilent MassHunter, respectively). We sought to identify a free alternative software for mass deconvolution to make our PG analysis pipeline accessible to everyone. MaxQuant was selected as a tool of choice since it represents a widely used software package to analyse high-resolution mass-spectrometric data for shotgun proteomics (*Cox and Mann, 2008*). MaxQuant has native support for Thermo MS data (RAW) and Sciex (WIFF) file formats and supports the open MS data format mzXML. Virtually any proprietary MS data file can be converted to the mzXML open format using freely available tools such as Proteowizard and TOPPAS, making this workflow universally applicable. As proof of concept, we converted *P. aeruginosa* data to an mzXML file (*Appendix 2—figure 1*), processed it using MaxQuant for mass deconvolution (*Appendix 2—figure 2*), and analysed it using PGFinder for PG structure and composition identification. We were able to identify all the expected muropeptides (*Table 2*). This result confirms that the automated analysis of PG datasets can be carried out using the MaxQuant freeware and our open-source script PGFinder (*Table 2* and *Table 2—source data 2*).

## Discussion

This study describes a workflow for the unbiased and automated analysis of bacterial PG using freely available resources. We analysed high-resolution MS datasets corresponding to PG fragments and demonstrated that this approach is a powerful tool to identify muropeptides and carry out comparative analyses based on the MS1 data.

MS analysis of bacterial PG has been carried out since the late 1980s (*Garcia-Bustos et al., 1988*; *Martin et al., 1987*). Whilst rp-HPLC-MS is now routinely used to explore PG structure, data analysis remains a manual process. Therefore, this step has become a bottleneck that prevents high-throughput analyses and introduces a series of issues regarding reproducibility. A major issue deals with the definition of the search space used since the list of muropeptides searched is not provided. Our previous work showed that an unbiased approach using shotgun proteomics tools identifies muropeptides containing unusual amino acids in *C. difficile* (*Bern et al., 2017*). Our unbiased search identified >106 masses matching *E. coli* muropeptide structures, representing a number strikingly larger than previously reported (*Kühner et al., 2014*). This level of complexity was anticipated based on the complement of enzymes involved in *E. coli* PG synthesis but had never been reported before. Our work therefore highlighted the limitations of search strategies reported so far, especially if we consider the fact that some bacterial PGs like *E. coli* have been extensively studied over the past 30 years.

Several issues and flaws associated with the manual analysis of PG MS data are addressed by our approach, which represents a robust, consistent, and open-access strategy. The PGFinder algorithm

**Table 2.** Automated identification of *P. aeruginosa* peptidoglycan fragments.

| Inferred structure | Mass | | Δppm | | MaxQuant |
| | Theoretical | Observed | This work | Anderson et al. | |
|---|---|---|---|---|---|
| GM (anhydro) | 478.1799 | 478.1780 | 4.0 | –2.7 | + |
| GM | 498.2061 | 498.2042 | 3.9 | –4.2 | + |
| GM (x2) (deacetyl) | 934.3755 | 934.3706 | 5.3 | –8.6 | + |
| GM (x2) (anhydro) | 956.3598 | 956.3551 | 5.0 | 6.0 | + |
| GM (x2) | 976.3860 | 976.3794 | 6.7 | –6.1 | + |
| GM (x3) (deacetyl) | 1412.5554 | 1412.5490 | 4.5 | –6.2 | + |
| GM (x3) (anhydro) | 1434.5397 | 1434.5348 | 3.4 | –7.5 | + |
| GM (x3) | 1454.5659 | 1454.5592 | 4.6 | –5.3 | + |
| GM (x4) | 1932.7458 | 1932.7352 | 5.5 | –5.1 | + |
| GM-AE (anhydro) | 678.2596 | 678.2567 | 4.3 | –9.1 | + |
| GM-AE | 698.2858 | 698.2830 | 3.9 | –12.9 | + |
| GM-AEJ (anhydro) | 850.3444 | 850.3401 | 5.1 | –10.6 | + |
| GM-AEJ | 870.3706 | 870.3676 | 3.5 | –5.9 | + |
| GM-AEJA (anhydro) | 921.3815 | 921.3765 | 5.4 | –9.9 | + |
| GM-AEJG | 927.3920 | 927.3868 | 5.6 | –8.9 | + |
| GM-AEJA | 941.4077 | 941.4045 | 3.4 | –5.0 | + |
| GM-AEJC | 973.3843 | 973.3763 | 8.2 | –2072.2 | + |
| GM-AEJL | 983.4593 | 983.4498 | 9.6 | –15.5 | + |
| GM-AEJK | 998.4703 | 998.4624 | 8.0 | –10.6 | + |
| GM-AEJM | 1001.4153 | 1001.4060 | 9.2 | –13.5 | + |
| GM-AEJAA | 1012.4448 | 1012.4413 | 3.4 | –7.8 | + |
| GM-AEJY (anhydro) | 1013.4091 | 1013.4242 | –14.9 | 17.8 | + |
| GM-AEJF | 1017.4433 | 1017.4347 | 8.4 | –15.0 | + |
| GM-AEJY | 1033.4353 | 1033.4278 | 7.2 | –5.3 | + |
| GM-AEJAV | 1040.4808 | 1040.4716 | 8.8 | –14.7 | + |
| GM-AEJIA | 1054.4964 | 1054.4874 | 8.5 | –11.3 | + |
| GM-AEJW | 1056.4394 | 1056.4455 | –5.8 | 4.0 | + |
| GM-AEJAM | 1072.4524 | 1072.4460 | 5.9 | –4.3 | + |
| GM-AEJKR | 1154.5667 | 1154.5631 | 3.1 | –8.1 | + |
| GM-GM-AE | 1176.4836 | 1176.4590 | 20.9 | –24.7 | + |
| GM-GM-AEJ | 1348.5684 | 1348.5457 | 16.9 | –24.9 | + |
| GM-GM-AEJA | 1419.6055 | 1419.5824 | 16.2 | –23.5 | + |
| GM-AEJA-GM-AEJ (amidase product) | 1313.5721 | 1313.5674 | 3.5 | –11.0 | + |
| GM-AEJA-GM-AEJA (amidase product) | 1384.6092 | 1384.6037 | 4.0 | –7.4 | + |
| GM-AEJ-GM-AEJ (anhydro) | 1702.7042 | 1702.6976 | 3.9 | 38.3 | + |
| GM-AEJ-GM-AEJ | 1722.7304 | 1722.7234 | 4.1 | –8.6 | + |
| GM-AEJA-GM-AEJ (double anhydro) | 1753.7151 | 1753.7043 | 6.2 | –7.2 | + |

*Table 2 continued on next page*

*Table 2 continued*

| Inferred structure | Mass | | Δppm | | |
| --- | --- | --- | --- | --- | --- |
| | Theoretical | Observed | This work | Anderson et al. | MaxQuant |
| GM-AEJA-GM-AEJ (anhydro) | 1773.7413 | 1773.7339 | 4.2 | −11.1 | + |
| GM-AEJA-GM-AEJ | 1793.7675 | 1793.7596 | 4.4 | −8.8 | + |
| GM-AEJA-GM-AEJA (dacetyl) | 1822.7941 | 1822.7808 | 7.3 | −7.4 | + |
| GM-AEJA-GM-AEJA (double anhydro) | 1824.7601 | 1824.7447 | 8.4 | −15.6 | + |
| GM-AEJA-GM-AEJA (anhydro) | 1844.7784 | 1844.7704 | 4.3 | −8.3 | + |
| GM-AEJA-GM-AEJG | 1850.7889 | 1850.8158 | −14.6 | 9.7 | + |
| GM-AEJA-GM-AEJA | 1864.8046 | 1864.7962 | 4.5 | −6.6 | + |
| GM-AEJA-GM-AEJK (anhydro) | 1901.8410 | 1901.8297 | 5.9 | −14.5 | + |
| GM-AEJA-GM-AEJL | 1906.8562 | 1906.8452 | 5.8 | −11.3 | + |
| GM-AEJA-GM-AEJK | 1921.8672 | 1921.8586 | 4.5 | −12.0 | + |
| GM-AEJA-GM-AEJF | 1940.8402 | 1940.8263 | 7.2 | −8.8 | + |
| GM-AEJA-GM-AEJY | 1956.8322 | 1956.8210 | 5.7 | −7.6 | + |
| GM-AEJA-GM-AEJAL | 1977.8933 | 1977.8813 | 6.0 | −10.7 | + |
| GM-AEJA-GM-AEJKR | 2077.9636 | 2077.9589 | 2.2 | −13.0 | + |
| GM-GM-AEJ-GM-AEJ | 2200.9282 | 2200.9000 | 12.8 | −17.7 | + |
| GM-GM-AEJA-GM-AEJ | 2271.9653 | 2271.9368 | 12.6 | −18.4 | + |
| GM-GM-AEJA-GM-AEJA | 2343.0024 | 2342.9734 | 12.4 | 411.4 | + |
| GM-AEJA-GM-AEJA-GM-AEJ (double anhydro) | 2677.1120 | 2677.1000 | 4.5 | −10.7 | + |
| GM-AEJA-GM-AEJA-GM-AEJ (anhydro) | 2697.1382 | 2697.1259 | 4.6 | −8.6 | + |
| GM-AEJA-GM-AEJA-GM-AEJ | 2717.1644 | 2717.1532 | 4.1 | −10.7 | + |
| GM-AEJA-GM-AEJA-GM-AEJA (double anhydro) | 2748.1491 | 2748.1363 | 4.7 | −11.0 | + |
| GM-AEJA-GM-AEJA-GM-AEJA (anhydro) | 2768.1753 | 2768.1674 | 2.9 | −11.2 | + |
| GM-AEJA-GM-AEJA-GM-AEJA | 2788.2015 | 2788.1919 | 3.4 | −9.7 | + |
| GM-AEJA-GM-AEJA-GM-AEJK (anhydro) | 2825.2379 | 2825.2205 | 6.1 | −9.3 | + |
| GM-GM-AEJA-GM-AEJA-GM-AEJ | 3195.3622 | 3195.3264 | 11.2 | −14.0 | + |
| GM-GM-AEJA-GM-AEJA-GM-AEJA | 3266.3993 | 3266.3630 | 11.1 | −12.5 | + |

Alternative structures were matched:
GM-AEJ-GM-AEJK.
GM-AEJ-GM-AEJKA (anhydro).
GM-AEJ-GM-AEJKA.
GM-AEJ-GM-AEJA-GM-AEJKA (anhydro).

The online version of this article includes the following source data for table 2:

**Source data 1.** *Pseudomonas aeruginosa* matched muropeptides not reported previously.

**Source data 2.** Raw output of automated search using MaxQuant and PGFinder.

has been designed to create dynamic databases that are ultimately combined to perform the final matching process. Optimisation of the search space relies on a preliminary identification of masses matching theoretical monoisotopic masses of monomers, limiting misidentifications based on mass coincidence. Another advantage is that the only information provided by the user is a restricted monomer database rather than a comprehensive one. This avoids a time-consuming operation, prone to human error. PGFinder uses XIC for quantification of PG fragments, providing high resolution, sensitivity, and reproducibility, as indicated by the comparisons across biological replicates (*Appendix 1— figure 2* and *Figure 4a*). It enables the accurate quantification of molecules with overlapping retention times and those present in very low abundance, with a large dynamic range (typically six orders of magnitude). Although a Matlab-based software package (Chromanalysis) has been described to automate the detection and quantification of UV peaks through Gaussian fitting (*Desmarais et al., 2015*), quantification using XIC is more straightforward. It is worth pointing out that the approach described here does not give absolute quantification of muropeptides. Instead, it allows a relative quantification of muropeptides. Nevertheless, this strategy remains suitable for comparative analyses and overcomes two major limitations associated with UV detection, namely detection threshold and co-elution of molecules.

Our *E. coli* PG analysis confirmed that PGFinder is a powerful tool that provides a much improved qualitative and quantitative PG analysis (*Kühner et al., 2014*; *Morè et al., 2019*). Combining an unbiased search with highly sensitive detection of individual structures is important for two reasons. Firstly, it opens the possibility to identify subtle modifications of PG structure, resulting from either a transient or a localised enzymatic activity such as that taking place at the septum. Secondly, it will permit the identification of previously undetected modifications that may provide new insights into our understanding of PG composition and dynamics. For example, we showed that *E. coli* PG contains a low abundance of deacetylated sugars. This observation is puzzling because no canonical PG deacetylase genes have been identified in this organism. Although the biological relevance of this property remains to be established, we cannot exclude the possibility that PG deacetylation in *E. coli* may contribute to PG homeostasis. Another striking outcome resulting from our automated search is the identification of a slightly higher amount of muropeptides containing anhydromuramic acid (4.55%) as compared to 2–3% (*Glauner et al., 1988*; *Liu et al., 2020*). To explain the discrepancy between our work and data from the literature, it is tempting to assume that most of the muropeptides containing anhydromuramic acid identified with PGFinder were simply not searched in previous studies. It is worth pointing out that none of the papers describing PG analysis published to date has reported the list of structures searched in the MS data analysed.

One of our objectives was to create an automated PG analysis tool accessible to the broadest audience possible, including people with no prior experience with programming or coding languages. Therefore, we shared PGFinder as a Jupyter Notebook allowing users to customise the search strategy depending on both the question asked and the instrument accuracy. PGFinder is particularly suitable for the characterisation of novel PGs with unknown composition or structural modifications and can be modified by users to add novel functionalities. However, a current limitation of this workflow is that it does not process MS/MS data. Therefore, the fragmentation spectra of individual monomers must be checked using dedicated tools to validate that the inferred structures are correct. We are currently working towards an integrated pipeline that includes MS/MS analysis to our PGFinder pipeline. The ability to disable some PG modifications means that the complexity of the search can be adjusted to focus on specific properties (e.g., the occurrence of acetylation/deacetylation, or amidation) or specific muropeptides resulting from lytic activities (e.g., unsubstituted MurNAc residues resulting from amidase activity). For PG that have already been well characterised (*E. coli*, *P. aeruginosa,* or *C. difficile*), the search parameters are already established, allowing a very straightforward analysis to be performed. Therefore, access to a custom, semi-quantitative sensitive analysis is ideal for comparing PG dynamics or differences in PG structure between a reference strain and isogenic mutants. Both reduced disaccharide peptides or lactyl-peptides (generated by beta-elimination) are identified using PGFinder.

We anticipate that an open access to PGFinder, in conjunction with freely available deconvolution tools, will allow researchers to carry out comparative MS1 analyses. The pipeline defined in this work enables reproducible and consistent data analysis. This represents the first step towards a standardised approach to PG analysis, opening the possibility to reanalyse datasets in repositories. The

modular structure of the open-source PGFinder code can be easily integrated into any specific work-flow for the automated processing of PG MS data.

# Materials and methods

**Key resources table**

| Reagent type (species) or resource | Designation | Source or reference | Identifiers | Additional information |
|---|---|---|---|---|
| Strain, strain background (*Escherichia coli*) | BW25113 | https://doi.org/10.1073/pnas.120163297 | RRID:Addgene_72340 | Model strain for PG analysis |
| Strain, strain background (*Clostridioides difficile*) | R20291 | https://doi.org/10.1128/JB.0073107 | | Model strain for PG analysis |
| Strain, strain background (*Clostridioides difficile*) | M7404 | https://doi.org/10.1371/journal.ppat.1002317 | | Model strain for PG analysis |
| Software, algorithm | PGFinder v.0.02 | This work | | Used for MS1 analysis of PG structure |
| Software, algorithm | Byos v.3.9–32 | Protein Metrics Inc | | Used for MS data deconvolution and MS/MS analysis |
| Software, algorithm | MaxQuant v2.0.1.0 | *Cox and Mann, 2008* | RRID:SCR_014485 | Used for MS data deconvolution |
| Software, algorithm | Perseus v.1.6.10.53 | *Tyanova et al., 2016* | RRID:SCR_015753 | Used statistical analysis of muropeptide abundance |

## Bacterial strains and culture conditions

*E. coli* BW25113 was grown at 37 °C in LB under agitation (250 rpm). *C. difficile* strains were cultured in heart infusion supplemented with yeast extract, L-cysteine, and glucose in an atmosphere of 10% $H_2$, 10% $CO_2$, and 80% $N_2$ at 37 °C in a Coy chamber or Don Whitley A300 anaerobic workstation.

## PG purification

PG was purified from exponential (*E. coli*) or late exponential (*C. difficile*) phase as described previously (*Eckert et al., 2006*; *Glauner, 1988*), freeze-dried, and resuspended in distilled water at a concentration of 5 mg/ml.

## Preparation of soluble muropeptides

PG (1 mg) was digested overnight with 25 µg of mutanolysin at 37 °C in 150 µl of 20 mM sodium phosphate buffer (pH 5.5). Soluble disaccharide peptides were recovered in the supernatant following centrifugation (20,000 × *g* for 20 min at 25 °C). To reduce muropeptides, equal volumes (200 µl) of the solution of disaccharide peptides and of borate buffer (250 mM, pH 9.0) were mixed. 2 ml of sodium borohydride was added, and the solution was incubated for 20 min at room temperature. The pH of the solution was adjusted to 4.0 with 20% orthophosphoric acid. Beta-elimination was carried out by mixing 200 µl of muropeptides with 64 µl of 32% (w/v) ammonia. After 5 hr at 37 C, the solution was neutralised with 60 µl of acetic acid glacial, freeze-dried, and resuspended in water (*Arbeloa et al., 2004*; *Eckert et al., 2006*).

The reduced muropeptides were desalted by reverse-phase high-performance liquid chromatography (rp-HPLC) on a C18 Hypersil Gold aQ column (3 µm, 2.1 × 200 mm; Thermo Fisher) at a flow rate of 0.4 ml/min. After 1 min in water-0.1% formic acid (v/v) (buffer A), muropeptides were eluted with a 6 min linear gradient to 95% acetonitrile-0.1% formic acid (v/v). Muropeptides were freeze-dried and resuspended in 100 µl. An aliquot of the desalted samples was analysed by rp-HPLC on the same column to measure the UV absorbance of the most abundant monomer (no isocratic step, muropeptides were eluted with a 30 min linear gradient to 15% acetonitrile-0.1% formic acid [v/v]). Samples were diluted to contain 150 mAU/µl of the major monomer and 10 µl were injected. Based on the dry weight of the PG sample, we estimated that this corresponded to approximately 50 µg of material.

## UHPLC-MS/MS

An Ultimate 3000 Ultra High-Performance Chromatography (UHPLC; Dionex/Thermo Fisher Scientific) system coupled with a high-resolution Q Exactive Focus mass spectrometer (Thermo Fisher Scientific)

was used for LC/HRMS analysis. Muropeptides were separated using a C18 analytical column (Hypersil Gold aQ, 1.9 µm particles, 150 × 2.1 mm; Thermo Fisher Scientific), column temperature at 50 °C. Muropeptides elution was performed by applying a mixture of solvent A (water, 0.1 % [v/v] formic acid) and solvent B (acetonitrile, 0.1 % [v/v] formic acid). After 10 µl sample injection, MS/MS data were acquired during a 40 min step gradient: 0–12.5% B for 25 min; 12.5–20% B for 5 min; held at 20% B for 5 min, and the column was re-equilibrated for 10 min under the initial conditions.

The Q Exactive Focus was operated under electrospray ionization (H-ESI II)-positive mode. Full scan (*m/z* 150–2250) used resolution 70,000 (FWHM) at *m/z* 200, with an automatic gain control (AGC) target of $1 \times 10^6$ ions and an automated maximum ion injection time (IT).

Data-dependent MS/MS were acquired on a 'Top 3' data-dependent mode using the following parameters: resolution 17,500; AGC $1 \times 10^5$ ions, maximum IT 50 ms, NCE 25 %, and a dynamic exclusion time 5 s.

## MS data deconvolution

Byos search parameters to get .ftrs file Protein Metrics Byos v.3.9–32 was used to identify and compute the XICs. The parameters used for mass deconvolution using MaxQuant v.2.0.1.0 are described in *Appendix 2—figure 2*.

## Data analysis

The crosslinking index and glycan chain length were calculated as described previously (*Glauner, 1988*). Label-free relative quantitation of muropeptides from triplicate *C. difficile* clinical isolates (R20291 and M7404) was performed using Byos 3.11, and statistical analysis of the quantitative data was performed using Perseus v. 1.6.10.53 (*Tyanova et al., 2016*). Briefly, muropeptide intensities were $\log_2$ transformed and normalised by subtraction of the median value. A two-sample Student's *t*-test was performed with a permutation-based FDR of 0.05 to determine statistically significant quantitative differences between the strains. Comparisons between R20291 and M7404 muropeptide distribution (mono-, di-, tri-, tetra-pentamer) was evaluated for statistical significance using GraphPad Prism (unpaired *t*-test).

## Runtime environment

Code is available at https://github.com/Mesnage-Org/PGFinder (*Patel, 2021*). https://github.com/Mesnage-Org/PGFinder/releases/tag/v0.02 will take you to the archived release used in this paper. We used Python 3 to write the MS1 package and demonstrate its functionality using demo scripts. PGFinder can be run through an interactive Jupyter Notebook hosted on mybinder for ease of use by those less familiar with Python code. A conda environment is provided to ensure reproducible execution. Regression testing has been implemented to ensure changes to code do not cause changes to important results. The GitHub contains an interactive version to run user's analysis and an end-to-end demo using samples data provided with the script (Interactive PGFinder). The sample data is a MaxQuant deconvolution output from the *E. coli* MS data analysed in the paper. The current version of the script can handle both .txt (MaxQuant) or .ftrs (Byos) deconvoluted data and offers the possibility for the user to include several modifications in the search. The time window for the 'clean up step' (in-source decay and salt adducts) as well as ppm tolerance for matching can also be defined by the user; the default values corresponding to these parameters used in this work are 0.5 min and 10 ppm.

## Data availability

All *E. coli* and *C. difficile* MS datasets generated in this study are available through the GlycoPOST repository (GPST000168; *Watanabe et al., 2021*). *P. aeruginosa* MS datasets are accessible via Figshare (*Anderson et al., 2020b*).

## Acknowledgements

We thank Dominique Mengin-Lecreulx (University Paris XI) for *E. coli* strain BW25113. Yong Kil and Eric Carslon (Protein Metrics) are acknowledged for their constant support. Motoshi Suzuki (NIH/NIAID) is acknowledged for insightful discussions.

## Additional information

### Competing interests

Andrew Nichols: Andrew Nichols is affiliated with Protein Metrics Inc. The author has no other competing interests to declare. Marshall Bern: Marshall Bern is affiliated with Protein Metrics Inc. The author has no other competing interests to declare. The other authors declare that no competing interests exist.

### Funding

| Funder | Grant reference number | Author |
| --- | --- | --- |
| Biotechnology and Biological Sciences Research Council | BB/M011151/1 | Ankur V Patel Stephane Mesnage |
| Medical Research Council | MR/S009272/1 | Stéphane Mesnage |
| Agence Nationale de la Recherche | ANR-10-LABX-62-IBEID | Ivo Gomperts Boneca Aline Rifflet |
| Agence Nationale de la Recherche | ANR-16-IFEC-0004 | Ivo Gomperts Boneca Aline Rifflet |
| Agence Nationale de la Recherche | ANR-18-CE15-0018 | Ivo Gomperts Boneca Aline Rifflet |
| Australian Research Council | DP210103374 | Dena Lyras Milena M Awad |

The funders had no role in study design, data collection and interpretation, or the decision to submit the work for publication.

### Author contributions

Ankur V Patel, Conceptualization, Data curation, Formal analysis, Investigation, Methodology, Resources, Software, Validation, Visualization, Writing – original draft, Writing – review and editing; Robert D Turner, Conceptualization, Data curation, Methodology, Software, Validation, Visualization, Writing – review and editing; Aline Rifflet, Investigation, Methodology, Writing – review and editing; Adelina E Acosta-Martin, Data curation, Formal analysis, Investigation, Methodology, Resources, Writing – review and editing; Andrew Nichols, Methodology, Resources, Supervision; Milena M Awad, Funding acquisition, Investigation, Supervision, Visualization, Writing – review and editing; Dena Lyras, Project administration, Supervision, Visualization, Writing – review and editing; Ivo Gomperts Boneca, Funding acquisition, Methodology, Resources, Writing – review and editing; Marshall Bern, Conceptualization, Funding acquisition, Methodology, Project administration, Software, Supervision, Validation, Writing – review and editing; Mark O Collins, Formal analysis, Funding acquisition, Methodology, Project administration, Resources, Supervision, Validation, Visualization, Writing – review and editing; Stéphane Mesnage, Conceptualization, Data curation, Formal analysis, Funding acquisition, Investigation, Methodology, Project administration, Resources, Supervision, Validation, Visualization, Writing – original draft, Writing – review and editing

### Author ORCIDs

Ankur V Patel http://orcid.org/0000-0001-8161-3455
Ivo Gomperts Boneca http://orcid.org/0000-0001-8122-509X
Mark O Collins http://orcid.org/0000-0002-7656-4975
Stéphane Mesnage http://orcid.org/0000-0003-1648-4890

### Decision letter and Author response

Decision letter https://doi.org/10.7554/eLife.70597.sa1
Author response https://doi.org/10.7554/eLife.70597.sa2

## Additional files

### Supplementary files
• Transparent reporting form
• Supplementary file 1. Step by step strategy for PG analysis.

### Data availability
All raw mass spectrometry data files have are available through the Glycopost repository (ref GPST000168).

The following previously published datasets were used:

| Author(s) | Year | Dataset title | Dataset URL | Database and Identifier |
|-----------|------|---------------|-------------|------------------------|
| Anderson EM, Sychantha D, Brewer D, Clarke AJ, Geddes-McAlister J, Khursigara CM | 2020 | Peptidoglycomics: Examining compositional changes in peptidoglycan between biofilm- and planktonic-derived *Pseudomonas aeruginosa* | dx.doi.org/10.6084/m9.figshare.10277909 | figshare, 10.6084/m9.figshare.10277909 |

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

## Appendix 1

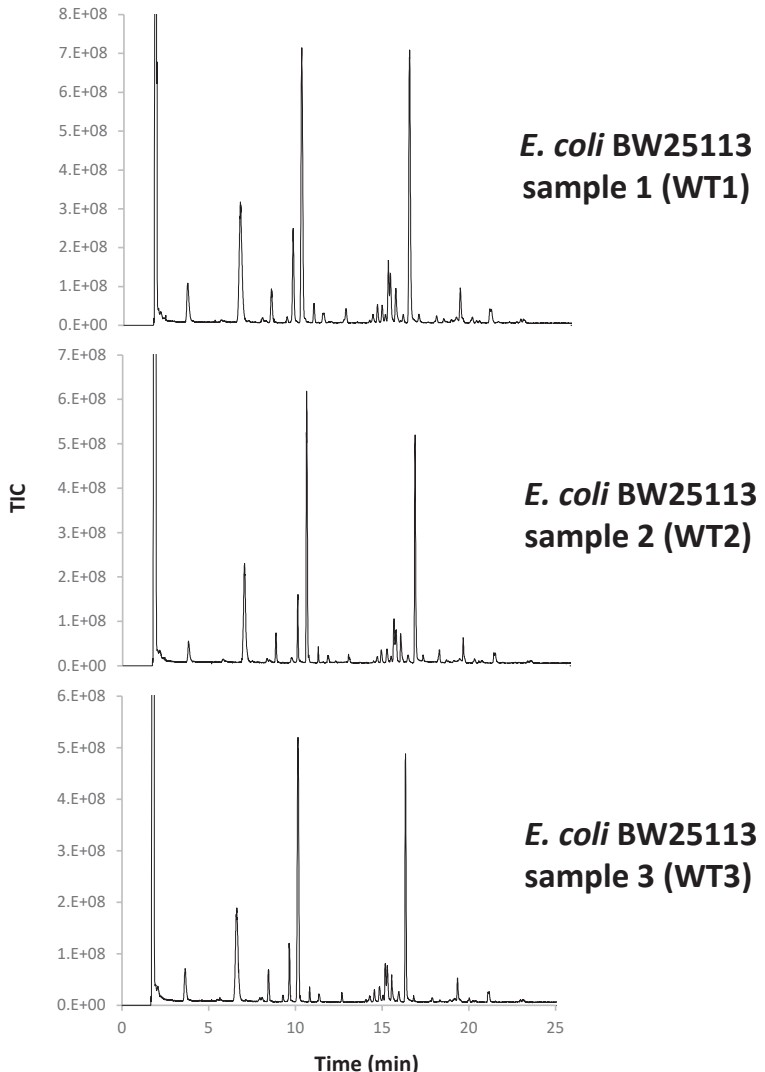

**Appendix 1—figure 1.** UHPLC-MS chromatogram of *E. coli* reduced disaccharide peptides.

**a**

|      | WT1   | WT2   | WT3   |
|------|-------|-------|-------|
| **WT1** | NA    | 0.974 | 0.973 |
| **WT2** | 0.974 | NA    | 0.960 |
| **WT3** | 0.973 | 0.960 | NA    |

**b**

|                     | WT1    | WT2    | WT3    | Av.    | SD    |
|---------------------|--------|--------|--------|--------|-------|
| **Glycan chains**   | 4.65%  | 3.78%  | 4.65%  | 4.36%  | 0.35% |
| **Monomers**        | 61.88% | 64.65% | 63.31% | 63.28% | 1.13% |
| **Dimers**          | 30.09% | 28.97% | 29.40% | 29.49% | 0.46% |
| **Trimers**         | 3.38%  | 2.60%  | 2.64%  | 2.87%  | 0.36% |
| Crosslinking index  | 16.16% | 15.34% | 15.57% | 15.69% | 0.34% |

**c**

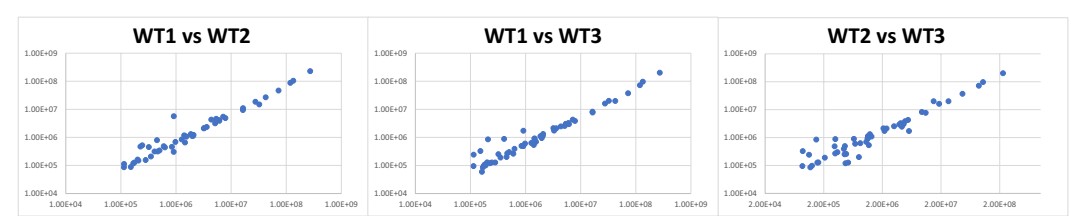

**Appendix 1—figure 2.** Consistency of *E. coli* PG analyses. (**a**) Pearson's correlation coefficients across biological replicates of *E. coli* BW25113. (**b**) Muropeptide distribution according to degree of crosslinking. The crosslinking index was calculated as described previously (*Glauner, 1988*). (**c**) Pairwise comparisons of intensities corresponding to individual muropeptides identified in biological replicates. WT1, WT2 and WT3 correspond to individual biological replicates; Av., average abundance; SD, standard deviation.

## Appendix 2

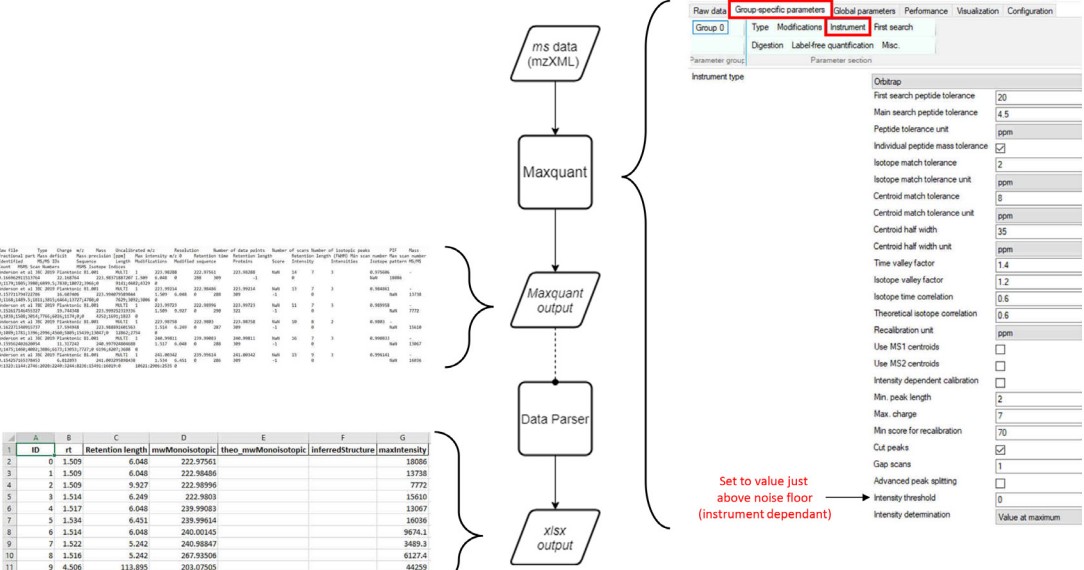

**Appendix 2—figure 1.** Workflow for production of MaxQuant compatible MS data files from Agilent QTOF data. Agilent MS data (data: .d) is converted by Proteowizard to a mzML format (data: XML). Relevant settings for Proteowizard are shown (left). mzML file is then converted by TOPPAS to a mzXML file (data: XML). Relevant settings are shown (right).

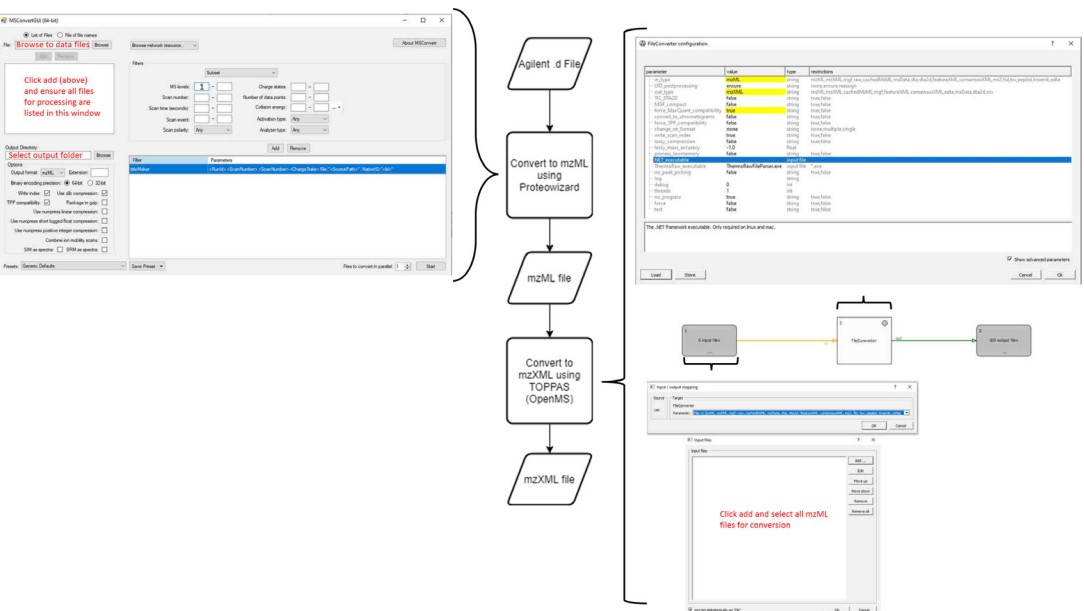

**Appendix 2—figure 2.** Workflow for MS data processing using MaxQuant, before automated analysis. mzXML (data: XML) is passed to MaxQuant (process) for deconvolution and monoisotopic mass determination. Default values used except where indicated (right). MaxQuant output (data: text file) is then passed to the data parser module (process). This module removes superfluous data and reformats remaining data to be compatible with the matching script as an Excel file (data: xlsx).

