## [Decision Letter]

**Acceptance summary:**

This manuscript presents the development and validation of a new tool for the characterization of peptidoglycan (PG), the essential cell wall polymer of bacteria. PG is a single large macromolecule that protects almost all bacterial cells. The newly developed open access tool will greatly facilitate comparative quantitative analyses and the determination of compositional diversity of PG, which might ultimately contribute to the development of new antibacterials that target this essential cell wall component.

**Decision letter after peer review:**

Thank you for submitting your article "PGfinder, a novel analysis pipeline for the consistent, reproducible and high-resolution structural analysis of bacterial peptidoglycans" for consideration by *eLife*. Your article has been reviewed by 3 peer reviewers, and the evaluation has been overseen by a Reviewing Editor and Gisela Storz as the Senior Editor. The following individuals involved in review of your submission have agreed to reveal their identity: Seemay Chou (Reviewer #2); Anthony Clarke (Reviewer #3).

Essential revisions:

The authors are encouraged to revise their manuscript by properly stating two weaknesses of their tool (as it stands in its current form):

1) The lack of important PG modifications in the mass spectrometry-based library of the muropeptides such as O-acetylation; inclusion of such modification would be required for the broader application of the analysis tool (e.g., as those modifications are frequently encountered in pathogenic Gram-positive bacteria);

2) The emphasis on comparative analyses between different samples (e.g., from the same species/strain but grown under different conditions) versus absolute quantification using MS, which might be more challenging. Future users are therefore encouraged to include appropriate controls.

*Reviewer #1 (Recommendations for the authors):*

Lines 285-288 say, "The present study confirmed that the complexity of bacterial PGs was greatly underestimated until now. Our unbiased search identified >106 masses matching *E. coli* muropeptide structures representing a number strikingly larger than previously reported (Kuhner, et al., 2014)." This may be an overestimate of the increase in complexity revealed by the new method of analysis. While older methods certainly identified fewer muropeptides, I think it was understood that the range of modifications and crosslinked dimers and trimers was greater than what was identified, and that many low-abundance muropeptides were present. Many of these could be predicted. This does not diminish the importance of this tool for allowing the rapid identification and quantification of many of these assumed muropeptides, but the actual increased demonstration of complexity should not be overstated.

Lines 298-301: The authors describe the advantages of muropeptide quantification using XIC, such as resolving overlapping peaks. There should also be a discussion of challenges of quantitative analysis using MS data, such as potential differences in ionization efficiency between muropeptides with significantly different peptide side chains, with possible lower detection of larger muropeptides due to lessor ionization, or differences of in-source fragmentation. Older methods using UV absorbance quantification were likely more accurate for quantification across the diversity of identified muropeptides, but missed low abundance muropeptides.

Lines 305-306: I agree that this represents a powerful tool for qualitative and reproducible PG analyses. The demonstration of absolute quantification is not clear. Yes, quantitative differences between the C. difficile strains is very clear, and the reproducibility of differences between the measured *E. coli* average cross-linking between this method and published values is clear, but whether the lower cross-linking determined by this method are more accurate is not clear. Is this due to greater detection of minor muropeptides or more accurate quantification of all muropeptides species? Or is the difference due to lessor detection of ions for the cross-linked muropeptides?

Lines 369-370 say, "Samples were diluted to contain 150mAU/μl of the major monomer. Based on the dry weight of the PG sample, we estimated that this corresponded to approximately 50μg of material" It is not clear what this 50 µg is. Does this dilution produce a suspension of 50 µg/µl? Does the resulting 10 µl injected for UHPLC contain 50 µg?

*Reviewer #2 (Recommendations for the authors):*

Below are several comments/questions on the paper as a whole but also on some specific sections.

1. The workflow was explained in detail in the text and also through the use of helpful schematics

2. Automating the analysis of MS cell wall data is a major advance in the field, especially through the use of open-source software.

3. Line 147, do you think the remaining 52-59% of total ion intensity is noise? If so, why so much? Or do you think there is some real signal that you are missing? If so, what could be improved to capture that?

4. Lines 149, what stereoisomers are you referring to? How do they form? Do you think these form in living cells?

5. Lines 158, 165, and 169, the authors found some discrepancies between current knowledge on *E. coli* PG and their data.

a. They stated in the text that further biochemical and molecular biology studies would be required to validate their findings.

6. Line 236, why did you increase the mass tolerance here to 25ppm? What was the problem if you used 10ppm as you did for the previous analysis?

7. The authors established the application of an alternative open-source deconvolution software.

a. They showed that it works as well as the commercial counterparts.

b. Lines 244-246, they described variability issues with the deconvolution step – what could be done to improve the software used here? This could help with the application of your approach across more datasets.

8. The authors provided multiple schematics that are helpful to visualize their workflow.

9. Their previous study and this one demonstrate the capacity for discovery when this type of unbiased approach was applied.

a. Especially regarding their novel findings for *E. coli* and C. difficile PG composition.

10. Their approach has advantages such as open-access and availability as a Jupyter notebook that would make it broadly available in the community. However, they also describe issues that need to be addressed.

a. One such issue is the ability to analyze MS/MS data.

11. The paper establishes high standards for other studies relying on MS data but also in general for other cell wall papers relying on chromatographic techniques

a. Biological replicates, list of searched muropeptides, and general data reporting

12. Some of the methods rely too heavily on previously published protocols. Could you provide brief details on the steps you followed in this study?

a. Lines 354, 361, and 395.

13. Check spellings of meso-diaminopimelic acid – it varies across the paper.

14. Correct g-D-Glutamate to γ-D-Glutamate – several instances across the paper.

*Reviewer #3 (Recommendations for the authors):*

1. Lines 125-126: Why did the authors limit the modifications to PG in their Library 3 to only those listed on these lines. Importantly, given that all pathogenic Gram-positive bacteria, and many Gram-negative bacteria produce O-acetylated PG, this reviewer is surprised that this modification was not (apparently) included, especially since reference is made to it on line included in Line 332 of the Discussion. This addition would be necessary for the analysis of the PG from these bacteria given the levels of O-acetylation which would greatly influence % compositional analyses of the various muropeptides.

2. Throughout the manuscript, including main and supplemental figures and tables, the authors have used J as their one-letter code abbreviation for diaminopimelic acid. Unfortunately, the IUPAC-IUB Joint Commission on Biochemical Nomenclature define J as the one-letter code for the combination of either isoleucine or leucine (I/L) ; please see, e.g., http://www.insdc.org/documents/feature_table.html#7.4.3.

All letters of the Latin alphabet are already used, and the Joint Commission states Dpm as the abbreviation for diaminopimelic acid. Using J as the (defined) abbreviation may have been acceptable with its isolated use, but it is very problematic when combined with the conventional code for the other amino acids. Clearly, using the three-letter abbreviation in combination with the one-letter code is awkward (particularly when convention usually requires the use of one or the other, not their combination) but in this instance, maybe the only option. Unless, the authors want to consider introducing another unconventional approach and move to eg., Greek and use delta for Dpm. This maybe going too far, but I can't think of any other alternative.

---

## [Author Response]

Essential revisions:The authors are encouraged to revise their manuscript by properly stating two weaknesses of their tool (as it stands in its current form):1) The lack of important PG modifications in the mass spectrometry-based library of the muropeptides such as O-acetylation; inclusion of such modification would be required for the broader application of the analysis tool (e.g., as those modifications are frequently encountered in pathogenic Gram-positive bacteria);

We agree that this was a limitation of the version of PGFinder we made available with the original submission. Rather than stating this weakness in the revised manuscript, we have added this functionality to PGFinder (now v.0.02; https://github.com/Mesnage-Org/PGFinder/releases/tag/v0.02). Users now have the possibility to search for O-acetylated PG fragments.

It is important to point out that given the complexity of PG structure, PGFinder will continue to be developed. We are very keen to receive feedback from users; the Jupyter notebook allows them to do so (see Author response image 1).

**Author response image 1. sa2fig1:** 

2) The emphasis on comparative analyses between different samples (e.g., from the same species/strain but grown under different conditions) versus absolute quantification using MS, which might be more challenging. Future users are therefore encouraged to include appropriate controls.

We agree with this point, but we suggest that relative quantification of muropeptides is sufficient for most experimental designs that researchers might wish to use. The quantification of, for example, proteomes or metabolomes by mass spectrometry is rarely absolute as relative quantification is sufficient to measure differences in the abundance of analytes across a wide range of sample types. Relative quantification, when performed with the appropriate number of replicates and robust statistical analysis, allows for confident measurement of changes in the abundance of muropeptides.

We have clearly stated that our data analysis pipeline is compatible with relative quantification in the revised manuscript (L. 304-307) and put emphasis on the fact that PGFinder is an ideal tool for comparative analyses, the most common type of PG analyses carried out to date.

Reviewer #1 (Recommendations for the authors):Lines 285-288 say, "The present study confirmed that the complexity of bacterial PGs was greatly underestimated until now. Our unbiased search identified >106 masses matching *E. coli* muropeptide structures representing a number strikingly larger than previously reported (Kuhner, et al., 2014)." This may be an overestimate of the increase in complexity revealed by the new method of analysis. While older methods certainly identified fewer muropeptides, I think it was understood that the range of modifications and crosslinked dimers and trimers was greater than what was identified, and that many low-abundance muropeptides were present. Many of these could be predicted. This does not diminish the importance of this tool for allowing the rapid identification and quantification of many of these assumed muropeptides, but the actual increased demonstration of complexity should not be overstated.

We agree with the reviewer’s comment. The point we tried to make was that PGFinder allows this complexity to be revealed, not that we discovered something conceptually novel. We have clearly stated this in the revised manuscript that the complexity described here was expected (L. 284-288).

Lines 298-301: The authors describe the advantages of muropeptide quantification using XIC, such as resolving overlapping peaks. There should also be a discussion of challenges of quantitative analysis using MS data, such as potential differences in ionization efficiency between muropeptides with significantly different peptide side chains, with possible lower detection of larger muropeptides due to lessor ionization, or differences of in-source fragmentation. Older methods using UV absorbance quantification were likely more accurate for quantification across the diversity of identified muropeptides, but missed low abundance muropeptides.

We discussed the limitations of our approach for quantitative analysis but insisted on the fact that the primary application of PGFinder is to carry out comparative analyses (L. 304-307). As we perform relative quantification of individual muropeptides across different samples, the relative ionisation of different muropeptides relative to each other is irrelevant. In-source fragmentation is handled by PGFinder and taken into account to consolidate intensities during the “clean up step”.

Lines 305-306: I agree that this represents a powerful tool for qualitative and reproducible PG analyses. The demonstration of absolute quantification is not clear. Yes, quantitative differences between the C. difficile strains is very clear, and the reproducibility of differences between the measured *E. coli* average cross-linking between this method and published values is clear, but whether the lower cross-linking determined by this method are more accurate is not clear. Is this due to greater detection of minor muropeptides or more accurate quantification of all muropeptides species? Or is the difference due to lessor detection of ions for the cross-linked muropeptides?

As stated previously, PGFinder is primarily a tool for comparative analyses.

The cross-linking index reported in this study is relatively low as compared to the one described in the paper cited. The seminal work of Glauner (1988) contains little information about the methods used to ascertain muropeptides identity. The interpretation of such discrepancies therefore seems highly speculative, and we would rather not try to address this point in the revised manuscript to avoid confusing the reader.

We feel that the issues with XIC quantification (highlighted in the revised version) could account for these differences.

Lines 369-370 say, "Samples were diluted to contain 150mAU/μl of the major monomer. Based on the dry weight of the PG sample, we estimated that this corresponded to approximately 50μg of material" It is not clear what this 50 µg is. Does this dilution produce a suspension of 50 µg/µl? Does the resulting 10 µl injected for UHPLC contain 50 µg?50µg of material was analysed by UHPLC-MS (L. 378-379).

Reviewer #2 (Recommendations for the authors):Below are several comments/questions on the paper as a whole but also on some specific sections.1. The workflow was explained in detail in the text and also through the use of helpful schematics.2. Automating the analysis of MS cell wall data is a major advance in the field, especially through the use of open-source software.

No comments required.

3. Line 147, do you think the remaining 52-59% of total ion intensity is noise? If so, why so much? Or do you think there is some real signal that you are missing? If so, what could be improved to capture that?

The PG extraction protocol we use does not involve a DNAse/RNase treatment or a step to break the cells. Our prior AFM experiments on sacculi prepared with this method clearly show that some material can be trapped inside the PG sacculi (doi:10.1111/j.1365-2958.2011.07871; DOI: 10.1038/ncomms2503). The enzyme used for the final digestion step is also present in the mixture.

4. Lines 149, what stereoisomers are you referring to? How do they form? Do you think these form in living cells?

The presence of stereoisomers has been reported in the literature (DOI: 10.1038/srep07494). These correspond to muropeptides that contain a small proportion of isomers that are incorporated in peptidoglycan peptide stems. Stereoisomers are therefore formed in living cells and are not an artefact associated with the chromatographic techniques used.

5. Lines 158, 165, and 169, the authors found some discrepancies between current knowledge on *E. coli* PG and their data.a. They stated in the text that further biochemical and molecular biology studies would be required to validate their findings.

No comment required.

6. Line 236, why did you increase the mass tolerance here to 25ppm? What was the problem if you used 10ppm as you did for the previous analysis?

We set the mass tolerance at 25 ppm to use the same threshold as described by Anderson et al. This comment has been added in the revised manuscript (L.235).

7. The authors established the application of an alternative open-source deconvolution software.a. They showed that it works as well as the commercial counterparts.b. Lines 244-246, they described variability issues with the deconvolution step – what could be done to improve the software used here? This could help with the application of your approach across more datasets

The two deconvolution software rely on distinct methods; Byos uses a machine learning algorithm to identify an isotopic series and then determine the charge state and m0 ion whereas MaxQuant uses a system based on constructing a graph network of the *m/z* spectra and identifying disconnected clusters in these networks to identify an isotopic series before determining charge state and m0 masses.

Neither software is open-source so we cannot improve them.

8. The authors provided multiple schematics that are helpful to visualize their workflow.9. Their previous study and this one demonstrate the capacity for discovery when this type of unbiased approach was applied.a. Especially regarding their novel findings for *E. coli* and C. difficile PG composition.

No comments required.

10. Their approach has advantages such as open-access and availability as a Jupyter notebook that would make it broadly available in the community. However, they also describe issues that need to be addressed.a. One such issue is the ability to analyze MS/MS data.

This point is discussed in the manuscript (L.330-333).

11. The paper establishes high standards for other studies relying on MS data but also in general for other cell wall papers relying on chromatographic techniquesa. Biological replicates, list of searched muropeptides, and general data reporting

No comment required.

12. Some of the methods rely too heavily on previously published protocols. Could you provide brief details on the steps you followed in this study?a. Lines 354, 361, and 395.

Reduction and β-elimination are briefly described (L. 364-370).

PG desalting is described in great details so no more details can be added!

The analysis using Perseus is widely used by proteomics users and all relevant details are provided. Many tutorials are available to explain the entire process in great detail.

13. Check spellings of meso-diaminopimelic acid – it varies across the paper.

Done.

14. Correct g-D-Glutamate to γ-D-Glutamate – several instances across the paper.

Done.

Reviewer #3 (Recommendations for the authors):1. Lines 125-126: Why did the authors limit the modifications to PG in their Library 3 to only those listed on these lines. Importantly, given that all pathogenic Gram-positive bacteria, and many Gram-negative bacteria produce O-acetylated PG, this reviewer is surprised that this modification was not (apparently) included, especially since reference is made to it on line included in Line 332 of the Discussion. This addition would be necessary for the analysis of the PG from these bacteria given the levels of O-acetylation which would greatly influence % compositional analyses of the various muropeptides.

We have modified PGFinder to include this modification as a search option.

2. Throughout the manuscript, including main and supplemental figures and tables, the authors have used J as their one-letter code abbreviation for diaminopimelic acid. Unfortunately, the IUPAC-IUB Joint Commission on Biochemical Nomenclature define J as the one-letter code for the combination of either isoleucine or leucine (I/L) ; please see, e.g., http://www.insdc.org/documents/feature_table.html#7.4.3.All letters of the Latin alphabet are already used, and the Joint Commission states Dpm as the abbreviation for diaminopimelic acid. Using J as the (defined) abbreviation may have been acceptable with its isolated use, but it is very problematic when combined with the conventional code for the other amino acids. Clearly, using the three-letter abbreviation in combination with the one-letter code is awkward (particularly when convention usually requires the use of one or the other, not their combination) but in this instance, maybe the only option. Unless, the authors want to consider introducing another unconventional approach and move to eg., Greek and use δ for Dpm. This maybe going too far, but I can't think of any other alternative.

Unfortunately, we cannot use Greek letters because the.csv files used by PGFinder (the muropeptide databases) do not support symbols or special characters. We argue that this is not a problem since the final output (the muropeptide table describing PG structure) can always be modified using the “search/replace” function. The letter J can therefore be replaced at the final stages of the PG analysis by any letter(s)/symbol.

To be consistent with the previous work (published by the reviewer) we have replaced the letter “J” by “m” in Table 1 (*E. coli* muropeptide table), Table 2 (*P. aeruginosa* muropeptide table), Figure 3 (Sankey diagram) and Figure 4 (*C. difficile* volcano plot). We added a sentence in the manuscript to explain this (L. 143-146).